# The Influence of Message Framing on Consumers’ Selection of Local Food

**DOI:** 10.3390/foods11091268

**Published:** 2022-04-27

**Authors:** Valentina Carfora, Maria Morandi, Patrizia Catellani

**Affiliations:** Department of Psychology, Catholic University of the Sacred Heart, 20123 Milan, Italy; maria.morandi@unicatt.it (M.M.); patrizia.catellani@unicatt.it (P.C.)

**Keywords:** environmental messages, local food, message framing, prefactual communication

## Abstract

Although local food purchasing provides several benefits to both consumers and producers, research on what recommendation messages can effectively promote the purchase of local food is still lacking. In the present study, 410 participants were involved in a 2-week intervention relying on prefactual (i.e., “If … then”) messages promoting the purchase of local food. All messages were sent through a research app to participants’ mobile phones and were focused on environmental consequences of purchasing (or not purchasing) local food. Four experimental conditions involving messages differing as to outcome sensitivity framing (i.e., gain, non-loss, non-gain and loss) were compared to a control condition. To test the effectiveness of the messages, before and after the 2-week intervention participants were involved in a choice task. They were asked to choose among fruits with different provenience, that is, from the participants’ municipality of residence or abroad. Results showed that all message frames increased the selection of local food, compared to control. Furthermore, pro-environmental consumers were more persuaded by messages formulated in terms of gains and non-gains, whereas healthy consumers were more persuaded by messages formulated in terms of losses or non-losses. Discussion focuses on the advantages of tailored communication to promote the purchase of local food.

## 1. Introduction

Although environmental deterioration may occur due to natural causes, human activities and interventions surely damage ecosystem health. A growing concern for sustainable development regards the domain of food production, which is responsible for around 26% of global annual greenhouse gas (GHG) emissions associated with climate change [1]. Given the aforementioned scenario, FAO and WHO underlined the need for a transition towards sustainable diets with lower environmental impact [2]. This transition can be pursued by promoting changes in numerous eating habits, such as completely or partially substituting meat and animal products with plant-based foods, and buying organic and/or local food.

So far, despite the increased interest in local food and the growing consumption of these products in many countries [3], only a little attention has been given to how to promote their purchasing. 

The scarce attention on the promotion of local food may be partly due to the absence of a common definition of it. The term “local food” has indeed various and sometimes contradictory definitions, but, in most cases, it is defined as food grown in close physical proximity to the consumer [4]. In the European context, the Joint Research Center of the European Commission defined the local food as food produced, processed, and retailed within a defined geographical area, approximately within a 20 to 100 km radius [5]. 

Another explanation for the limited attention to local food promotion may be related to the ongoing debate on the environmental sustainability of local food supply chains. According to some research results, local food supply chains have a low environmental impact, as they reduce greenhouse gas emissions associated with food transportation. Moreover, the practices of many producers in local food systems (e.g., crop rotation, creation of field borders to provide a refuge for native biodiversity, packaging reduction, or moderation in the use of fertilizers and chemicals) contribute to reducing the ecological footprint of food production [6,7]. According to other studies, local food supply chains may have a higher environmental impact than non-local food supply chains, for example when local food is cultivated in heated glasshouses [8,9]. This debate highlights the need to consider the complexity of food supply chains, whose environmental impact is not determined only by the distance traveled by food, but also by production methods and processes’ efficiency [9,10]. For this reason, in the present study participants were presented with the environmental benefits of local food considering not only the geographical proximity but also the sustainable production/distribution methods adopted by the local farmers who do not follow the large-scale distribution logic.

To fill the gap in the scientific literature about how to promote the consumption of local food, in the present study, we tested the effectiveness of a 2-week intervention consisting in sending daily messages via an app on the environmental consequences of local food consumption. In this intervention, we employed prefactual (i.e., “If … then”) messages [11] and compared the effectiveness of four message frames, differing in how they presented the environmental consequences of local food consumption. Following the model of outcome sensitivities [12], we made a distinction among gain, non-loss, non-gain and loss messages, and tested whether they would differently persuade consumers to select local food. Then, we evaluated to what extent the four message frames were effective as a function of the receivers’ health and environmental drivers in food choice.

## 2. Theoretical Background

### 2.1. Persuasive Messages to Promote Sustainable Food Purchase 

Sustainable food communication can be formulated using diverse linguistic styles, contents, and framing. Considering the linguistic style, some researchers agree that a prefactual formulation of the information (i.e., “If … then”), which presents a hypothetical future outcome of present actions, makes messages more effective in influencing receivers’ intentions and behaviors related to sustainable food choices, compared to the factual formulation, which illustrates the causal relationship between a behavior and its outcome [11]. 

As regards the message contents to promote sustainable eating habits, past scholars have commonly referred to animal welfare [13], human wellbeing [11], health [11,13,14,15,16] and environment [13,14,15,16,17,18,19]. In the present study, we focused on environmental content due to the environmental benefits associated with local food production. Generally, past research on the effectiveness of messages focused on the environmental consequences of sustainable eating habits has offered mixed results. Some researchers found that the environmental messages did not influence consumers’ evaluation of meat [13], and had a limited impact on the intention to reduce meat consumption or follow a sustainable diet [14,15]. Instead, other scholars showed that these messages increased the intention to reduce meat consumption [16]. In the case of the promotion of local food purchasing, only one study analyzed the effectiveness of messages, which were focused on local farmers’ support, high quality, and healthiness of local food, showing that all these messages failed in enhancing consumers’ positive attitude towards local food purchasing [20]. However, no previous studies tested the persuasiveness of environmental messages. 

Considering message framing, which refers to the fact that a given message content can be presented in different, although objectively equivalent, versions [21,22,23], many researchers have shown that message effectiveness may vary according to the positive or negative valence of the envisaged outcomes. More precisely, a recommendation message can be framed either with a positive valence, by describing the positive consequences of the adherence to the recommendation (i.e., positive frame; e.g., “If you buy local food, you contribute to the safeguard of biodiversity”), or with a negative valence, by describing the negative consequences of the non-adherence to the recommendation (i.e., negative frame; e.g., “If you don’t buy local food, you contribute to the loss of biodiversity”) [24,25]. The framing of the outcome valence has been largely applied to health messages, showing that positively framed health messages are more effective in promoting preventive behaviors, while negatively framed messages are more effective for detecting behaviors [24,26]. However, its application to environmental messages is still scarce and the results have not been univocal. On the one hand, some researchers found that positively framed environmental messages lead to higher pro-environmental intentions and corresponding behaviors, compared to negatively framed messages [17,27,28,29]. On the other hand, some researchers found that individuals exposed to negatively framed environmental messages are more likely to act a pro-environmental behavior [30]. Still, other researchers found that message framing alone did not influence environmental behaviors [31,32,33].

These mixed results of past research on the effects of valence framing can be attributed, at least in part, to the fact that the distinction between positively and negatively framed messages is not fine-grained enough [34,35]. In this vein, Cesario and colleagues [12] proposed a more fine-grained classification of valence framing, defined as the *outcome sensitivity level* of message framing. According to this classification, the expected outcomes presented in the messages can be formulated in four different ways: (a) *gain* messages focus on the presence of positive outcomes (e.g., “If you buy local food, you contribute to the safeguard of biodiversity”); (b) *non-loss* messages focus on the absence of negative outcomes (e.g., “If you buy local food, you avoid contributing to the loss of biodiversity”); (c) *non-gain* messages focus on the absence of positive outcomes (e.g., “If you don’t buy local food, you miss the opportunity to contribute to the safeguard of biodiversity”); and (d) *loss* messages focus on the presence of negative outcomes (e.g., “If you don’t buy local food, you contribute to the loss of biodiversity”). 

Some previous research on the effect of the outcome sensitivity level of message framing found that messages focused on present outcomes (i.e., gain and loss) are more persuasive than messages focused on absent outcomes (i.e., non-gain and non-loss) [34,35,36]. However, as regards environmental messages, only a few studies have tested the effect of the outcome sensitivity level of framing, showing that the effectiveness of the four message frames only varies according to the receivers’ psychosocial characteristics, such as their prior beliefs [37].

### 2.2. The Moderating Role of Consumers’ Characteristics 

A long tradition of communication research showed that message persuasiveness is influenced by the characteristics of the recipients [38]. In line with the abovementioned results, in this study, we expected that environmental messages would influence receivers according to their food choice drivers. Particularly, we have considered the relevance they attribute to environmental protection and health as key determinants of their reactions to environmental messages. People interested in environmental protection tend to carefully evaluate the environmental impact of their food choices; thus, they can be strongly motivated by pro-environmental information. However, people guided by a health interest, who tend to choose eating habits that guarantee their physical wellbeing [39], can also be persuaded by environmental information on their food choices [40]. This is because people are increasingly recognizing a close connection between humanity and nature and, in turn, between environmental damages and their health, as suggested by the new human interdependence paradigm and the evidence that people engaged in either a transition toward a more sustainable diet or its maintenance have high levels of both environmental and health concerns [40,41,42]. Within the promotion of local food purchasing, place identity is another possible consumer characteristic that may influence message persuasiveness.

## 3. The Present Study

Based on the above-discussed literature, in the present study, we tested whether exposure to prefactual environmental messages differing as to the outcome sensitivity level of message framing would influence recipients’ selection of local food in a choice task. 

Consistent with previous results on the effectiveness of the exposure to prefactual messages focused on the environment [16], we expected that participants reading these messages would increase their selection of local food, as compared to participants assigned to a control condition with no exposure to prefactual messages (H1). 

As to the effect of the outcome sensitivity level of message framing, as well as the possible moderation of environmental and health drivers, we did not formulate any specific hypothesis, but only two research questions in consideration of the absence of previous results on the promotion of local food using differently framed messages. We first assessed whether the effect of message exposure on food choice would vary according to the outcome sensitivity level of message framing (i.e., gain, non-loss, non-gain and loss) (RQ1). 

Then, in line with studies emphasizing the central role of health and environmental drivers in sustainable eating and local food consumption [14,39], we assessed whether the effect of environmental messages about local food consumption consequences would be moderated by the recipient’s environmental and health drivers (RQ2).

## 4. Materials and Methods

### 4.1. Sample and Procedure

Ethical approval for this study was provided by the Catholic University of the Sacred Heart. Using GPower 3.1, we conducted a sample size estimation considering an Effect size f = 0.25. With an alpha = 0.05, power = 0.80, number of groups = 5 (message conditions), number of measurements = 2 (1 measure at 2 time points) and *p* = 0.05 the projected sample size needed was approximately N = 196, and specifically about 39 participants per group. To ensure this sample size despite any dropouts during the intervention phase, we aimed to at least double the number of participants needed. 

In May 2021, we invited 450 individuals to participate in this study through Prolific (https://www.prolific.com, accessed on 10 March 2022); a platform for online subject recruitment designed for research. Prolific verifies the identity of all the subjects asking to enroll as participants, as well as collects their socio-demographic information through which it defines the population available for the studies. While setting a study on Prolific, researchers are informed about the available population and they can pre-screen participants and define exclusion and inclusion criteria. Prolific offers the possibility to conduct one-time research, as well as longitudinal research. All participants are explicitly informed that they are recruited for participation in research, and on the Prolific webpage, they can easily find the studies for which they are suitable, based on the criteria defined by the researchers. Before they agree to take part in the study, participants are also informed about the expected payment for participation in the study. To ensure fair pay for participants, Prolific defines a fixed minimum payment based on the time required to complete the study. Once they correctly completed the study, participants are paid by the researcher through Prolific. In the present study, participants were expected to be residents of Italy and have a Prolific record of at least 75% satisfactorily completed experiments. The study was advertised as research on purchasing behavior, about a total of 40 min in length, and distributed in 2 weeks. Those completing the entire research were paid £5.00. 

After accessing the study on Prolific, participants provided informed consent through a questionnaire implemented through the Qualtrics platform. In the same questionnaire, participants found the instructions to access the PsyMe (PsyMe is a free smartphone app of the Catholic University of the Sacred Heart, developed to support scientific research in the field of social psychology and artificial intelligence (https://apps.apple.com/it/app/psyme/id1536587889; https://play.google.com/store/apps/details?id=uncatt.unipv.xtream.psyme&hl=en_US&gl=US, accessed on 10 March 2022). The PsyMe app respects participants’ privacy and anonymity thanks to the assignment of an anonymous code to each participant. The current version of the PsyMe app allows sending questionnaires, messages and push notifications to remind message reading using an anonymous alphanumerical code, and to correctly participate in the study using the app. The alphanumerical codes were generated using an automatic randomization sequence, through which participants were randomly assigned to one of the five experimental conditions of the study (see Section 4.3 below).

Using the PsyMe app, participants access the Time 1 (T1) questionnaire. At the beginning of the questionnaire, we provided participants with instructions to properly fill out the online questionnaire. Then, participants received one message a day for 14 days, except for participants assigned to the control condition. At the end of the 14-day intervention period, all participants completed the Time 2 (T2) questionnaire. Finally, participants received feedback on the aims of the study. A control question to verify if participants’ replies were reliable was included in both the questionnaires. 

Figure 1 shows participants’ flow during the study. At T1, 410 participants accessed the PsyMe app and correctly completed the questionnaire (189 females, 218 males, 3 non-binaries; age range 18–64 years, *M* age = 26.83, *SD* age = 7.62). After the intervention, at T2, 353 participants correctly filled in the second questionnaire and were retained as the final sample of our study. Among the excluded participants, 7 failed the control question at T1 and 2 at T2.

### 4.2. Measures at Time 1 (T1)

The questionnaire at T1 included several measures. Below, we report the relevant measures for the present paper.

First, we asked participants socio-demographic information including age, sex, level of education, marital status, monthly income, region, municipality of residence and its demographic dimension.

*Environmental driver.* Participants rated the relevance of environmental drivers when buying groceries on a Likert scale ranging from “not relevant at all” (1) to “very relevant” (5) (“To what extent do environmental impact usually guide you in the choice of food to buy?”) [43].

*Health driver.* Participants rated the relevance of health drivers when buying groceries on a Likert scale ranging from “not relevant at all” (1) to “very relevant” (5) (“To what extent do health impact usually guide you in the choice of food to buy?”) [43].

*Choice task related to the selection of local food.* To measure the selection of local food, participants were asked to perform a choice task in which they had to choose five fruits from a list of ten (see Table A1). Together with vegetables, fruit is one of the most sold local food products [3]. Fruit and vegetable consumption is also determined by the same food choice drivers, specifically health and environmental concerns [44]. In this study, we only proposed a fruit selection task to guarantee the feasibility of the experiment, which otherwise might be too time consuming for participants and would make them experience cognitive overloading. The instructions were as follows: “Imagine buying fruit online. Please, choose five among the following ten types of fruit”. The task randomly presented five fruits produced in places located more than 100 km away from the municipality of residence of the participant and five fruits produced in places within 100 km from the municipality of residence. All fruits were depicted and described by name, weight, size, category, origin, date of packaging and price. The information on the product’s origin was deliberately inserted among other information so that it did not stand out too much. For the same reason, participants were not provided with a budget of expenditure to not make the price information more salient and relevant compared to the others.

*Place identity related to municipality of residence.* Since we identified as local the foods coming from areas adjacent to the place of residence, we controlled for the potential effect of the participants’ place identity. Place identity refers to the emotional attachment to a specific place, especially the place where one lives, which is a repository of meaningful emotions and relationships [45]. Participants’ place identity related to their municipality of residence was measured with six items on a 7-point Likert scale (e.g., “I feel [municipality of residence] is part of me … strongly disagree (1)–strongly agree (7)”; *α* = 0.96) [43]. A Qualtrics feature ensured that participants read the name of the previously declared municipality of residence.

### 4.3. Message Intervention

During the 14-day intervention (between T1 and T2) all participants received daily persuasive messages via the PsyMe app. Thus, fourteen messages were sent in each condition. The full list of messages is reported in Table A2. All messages were focused on the environmental consequences of purchasing or not purchasing local food. Moreover, they were formulated in prefactual style, which consists of a conditional proposition about an action–outcome linkage that may (or may not) occur in the future (e.g., “If I take action X, it will lead to outcome Y”) [11]. Participants in the *gain message* condition received messages focused on the positive environmental consequences of purchasing local food (e.g., “Buying food produced in places close to us favors the survival of local agricultural varieties. If you buy local food, you contribute to safeguarding biodiversity”). Participants in the *non-loss message* condition received messages on the avoidance of negative environmental consequences of purchasing local food (e.g., “Buying food produced in places close to us avoids the disappearance of local agricultural varieties. If you buy local food, you avoid contributing to the loss of biodiversity”). Participants in the *non-gain message* condition received messages on the loss of positive environmental consequences of not purchasing local food (e.g., “Buying food produced in places far from us hinders the survival of local agricultural varieties. If you don’t buy local food, you miss the opportunity to contribute to the safeguard of biodiversity”). Participants in the *loss message* condition received messages on the negative environmental consequences of not purchasing local food (e.g., “Buying food produced in places far from us favors the disappearance of local agricultural varieties. If you don’t buy local food, you are contributing to the loss of biodiversity”).

### 4.4. Measures at Time 2 (T2)

In the questionnaire at T2, we measured again the selection of local food, using the task employed at T1 (Table A1). We also included some scales to assess participants’ evaluation of the messages received through the PsyMe app.

*Message reading frequency* was obtained through the PsyMe app, which keeps track of the reception of the messages.

*Manipulation check* was conducted by asking the participants to select among four messages the one that was most similar to the messages they received for 14 days through the PsyMe app.

*Message tone* was assessed with an item using a semantic differential scale ranging from “extremely negative” (1) to “extremely positive” (7): “How would you rate the tone of the information presented in the messages?” Higher values indicated a more positive perception of the information tone [46].

*Message involvement* was measured with three items using a Likert scale ranging from “completely disagree” (1) to “completely agree” (7): “Messages got me involved in what they had to say”; “Messages were interesting”; “Messages seemed relevant to me” [46]. Higher values indicated a higher involvement with the messages (*α* = 0.86).

*Message trust* was assessed with three items on a Likert scale ranging from “not at all” (1) to “completely” (7): “The information is credible”; The information is reliable”; “The information was truthful” [46]. Higher values indicated higher trust in the messages (*α* = 0.92).

*Systematic processing* was assessed with five items on a Likert scale ranging from “not at all” (1) to “completely” (7): “While I was reading messages, … I thought about what actions I might take based on what I read; I found myself making connections between the information and what I’ve read or heard about elsewhere; I thought about how the information related to other things I know; I tried to think about the importance of the information for my daily life; I tried to relate the ideas in the information to my health” [47]. Higher values indicated higher systematic processing of the messages (*α* = 0.87).

### 4.5. Data Analysis

All analyses were conducted with SPSS 23. As preliminary analyses, we first ran descriptive and correlation analyses to explore the measured variables and the relationships among them. We then checked for the absence of biases in randomization and dropouts using MANOVAs and Chi-square tests. Next, we ran an ANOVA to check if the message reading frequency was influenced by the message frame. As for the manipulation check, with Chi-square and ANOVA tests we verified if participants correctly recognized the differences among message frames. Then, we ran a MANOVA to check if differently framed messages were perceived as equally involving, credible, and were equally systematically processed. 

As to the main analyses, we used repeated measures ANOVAs and *t*-test to investigate if the messaging intervention (H1) and the message framing (RQ1) were effective in enhancing the selection of local food in the choice task. Finally, to investigate if the effectiveness of message framing was influenced by the recipients’ environmental and health drivers (RQ2), we ran a moderation analysis using Model 2 of the PROCESS macro for SPSS [48].

## 5. Results

### 5.1. Preliminary Analyses

Table 1 reports the socio-demographic characteristics of the final sample. The sample was well balanced in terms of gender. However, most participants were single, young or young adults, with a high school diploma, residents in Northern Italy and with a monthly income below EUR 1200.

When buying food, our participants gave greater importance to the health impact of their choices. After the message exposure, instead, the choice task was also correlated with the environmental driver. Table 2 reports the means and standard deviation of place identity, environmental and health drivers, and choice tasks both in the total sample and among conditions. Table 3 reports the correlations between these variables. 

To check if randomization was successful, we used a multivariate analysis of variance (MANOVA), testing if there were differences among conditions on environmental and health drivers, choice task, place identity and age at T1. Results did not show any significant main effect of message conditions (*p* = 0.71, *ηp*^2^ = 0.01) on these variables. Chi-square also did not show any significant differences in gender, marital status, level of education, monthly income, region of residence and the dimension of the municipality across different conditions (all *p* > 0.11). This suggests that randomization was adequate, with the five conditions being comparable to the baseline variables. 

Regarding dropouts (Figure 1), 51 participants dropped out at post-test (T2). Chi-square did not show any significant differences in dropouts (*p* = 0.98) across different conditions. Moreover, a MANOVA analysis indicated that dropouts were neither explained by place identity nor environmental and health drivers (*p* = 0.93). These results and the low rate of dropout in our 14-day intervention allowed us to assert that dropouts were not determined by conditions or participants’ environmental and health drivers, and place identity.

### 5.2. Message Evaluation

First, we verified if participants in the experimental conditions read frequently our messages. We found that 19% of subjects read them every 2–3 days, and 76.2% read them every day. Then, we checked if the message reading frequency was influenced by the message frame using an ANOVA, which did not reveal any difference among conditions (*p* = 0.75). Next, we conducted a manipulation check by controlling whether subjects correctly identified the message frame they received during the intervention. Chi-square showed significant differences in message frame identification (179.05; *p* = 0.001) across different conditions, confirming that the difference among message frames was understood and recognized by participants. Moreover, we checked if there were differences in the evaluation of the tone of the messages in the four message conditions using an ANOVA. Results showed a main effect of condition (*F* (1282) = 8.16 *p* > 0.001, *ηp*^2^ > 0.08), thus participants differently perceived the tone of the four message frames. Specifically, pairwise comparisons (*p* > 0.05) indicated that participants perceived the loss messages (*M* = 4.54; *SD* = 1.54) as more negative compared to the gain (*M* = 5.60; *SD* = 1.33; *p* = 0.001), non-loss (*M* = 5.42; *SD* = 1.09; *p* = 0.001) and non-gain messages (*M* = 5.08; *SD* = 1.45; *p* = 0.05). Furthermore, we analyzed whether there were differences among conditions in participants’ message involvement (*M* = 5.33; *SD* = 1.14), message trust (*M* = 5.23; *SD* = 1.03) and message processing (*M* = 5.04; *SD* = 1.14). MANOVA results showed that all messages were perceived as equally involving and credible, and all of them were systematically processed (*p >* 0.93).

### 5.3. Effects of Message Intervention on Choice Task

We then examined whether message intervention changed the selection of local fruit (H1) and whether there was a different effect of message frames (RQ1). First, we conducted a 5 (control condition and four message conditions) × 2 (T1 vs. T2) ANOVA with choice task related to local fruit as dependent variable, with repeated measures on the last factor (Table 4). Results showed a great main effect of time effect on choice task. *t*-test comparisons, *t* = 7.70; *df* = 350; *p* = 0.001; *95% CI*: 0.30, 0.52; *Cohen’s d* = 0.5, showed that participants selected more local fruit at T2 (*M* = 3.34; *SD* = 1.16) than at T1 (*M* = 2.88; *SD* = 1.04). We did not find a significant condition effect. However, the interaction between message condition and time was significant. Therefore, there were differences in the extent to which interventions resulted in a greater choice of local fruit in T2 as compared to T1. Specifically, from T1 to T2 participants in the gain (*M* = 3.63; *SD* = 1.13; *p* = 0.003), non-loss (*M* = 3.48; *SD* = 1.13; *p* = 0.023), non-gain (*M* = 3.51; *SD* = 1.15; *p* = 0.017) and loss message conditions (*M* = 3.50; *SD* = 1.16; *p* = 0.019) increased the selection of local fruit compared to participants in the control condition (*M* = 3.04; *SD* = 1.17). 

Then, we controlled whether our results were or not independent of the participants’ environmental driver, health driver and place identity. Thus, we ran the same repeated measure ANOVA including health driver, environmental driver and place identity as covariates. (Appendix B). This analysis confirmed an interaction effect between time and condition, supporting that from T1 to T2, participants in the message conditions increased their choice of local fruit, compared to participants in the control condition.

In sum, these results supported our H1, showing that the provision of information on the environmental consequences of local fruit purchasing was effective in increasing the selection of these foods in our choice task, independently from the participants’ baseline environmental driver, health driver, and place identity. However, regarding our RQ1, we did not find any significant difference among the message frames. Thus, our messages were effective regardless of the environmental consequences formulation.

### 5.4. The Moderating Role of Environmental and Health Drivers on Message Framing Effectiveness

To address our RQ2, we explored whether the effects of the four message frames depended on the strength of participants’ environmental and health drivers. For this analysis, we calculated the change in the choice task as a difference score between the local fruit choice at T2 and the choice made at T1. Then, we ran a moderation model with message condition as the independent categorical variable, environmental and health drivers as the moderators, and the change in the choice task as the dependent variable (Model 2 of the PROCESS macro for SPSS; Table 5) [48]. Neither environmental nor health drivers had a main effect on change in the choice task. This result indicated that receivers’ environmental and health drivers did not influence their selection of local fruit, independently from the message frame to which participants were exposed. Instead, results of the interaction effects showed that participants in the gain and non-gain message conditions increased their selection of local fruit as their environmental driver increased. Differently, participants in the non-loss and loss message conditions increased their selection of local fruit as their health driver increased.

These results showed that the stronger was the consumers’ environmental driver, the more they were sensitive to messages framed in terms of presence or absence of environmental gains, whereas the stronger the consumers’ health driver, the more they were sensitive to messages framed in terms of presence or absence of environmental losses. In the following sections, we specify how the consumers’ health and environmental drivers interacted with message conditions on changing the task choice.

#### 5.4.1. Participants with a Weak Environmental Driver in Food Choice

Conditional effects of the message conditions at a low level of environmental and health drivers were not significant, indicating that participants with weak environmental and health drivers were indifferent to the message frame. Instead, at a low level of environmental and a medium or a high level of health driver, the loss message condition had a significant conditional effect. Hence, participants disinterested in the environmental impact of their food choices, but with a medium or high level of health driver, were persuaded by messages focused on the environmental damage deriving from the non-consumption of local products (Figure 2; Table 6).

#### 5.4.2. Participants with a Medium Environmental Driver in Food Choice

At a medium level of environmental and a low level of health driver, gain and non-gain message conditions had a conditional effect on food choice. Participants who were only moderately concerned about the impact of food on the environment were persuaded by messages focused on the presence or absence of environmental gains deriving from the consumption (or non-consumption) of local food (Figure 2; Table 7).

At a medium level of environmental and health drivers, all message conditions effects were conditioned by the moderators. The most persuasive condition was exposure to the loss messages, followed by exposure to the gain, then the non-loss, and finally the non-gain messages. Participants who were moderately interested in the environmental and health consequences of their food choices were convinced by all types of message frames, even if they preferred simpler (gain, loss) than more complex (non-loss, non-gain) messages (Figure 2; Table 7).

At a medium level of environmental driver and a high level of health driver, the conditional effects of gain, non-loss and loss message conditions were significant. In this case, the most effective message condition was the loss, followed by the non-loss, and the gain message conditions. Therefore, the consumers who were moderately interested in the environmental effects and strongly interested in the health consequences of their food choices were more influenced by messages focusing on the presence or absence of the environmental risks derived from not purchasing local food. However, they were also motivated by messages focused on the environmental benefits of local food consumption (Figure 2; Table 7).

#### 5.4.3. Participants with a Strong Environmental Driver in Food Choice

At a high level of environmental driver and a low level of health driver the most convincing message was the non-gain, followed by the gain messages. Thus, participants who were strongly interested in the environmental consequences, but disinterested in the health consequences of their food choices, were persuaded by messages focused on the presence or absence of the environmental benefits of purchasing (or not) local food (Figure 2; Table 8).

At a high level of environmental drivers and a medium level of health drivers, all message conditions had a significant conditional effect. The most persuasive condition was the exposure to gain messages, followed by exposure to non-gain, non-loss and loss messages. Therefore, participants who were strongly interested in the environmental impact and moderately interested in the health impact of their food choices were sensitive to all message frames, even if they preferred those emphasizing the presence or absence of the environmental gains derived by purchasing (or not) local food (Figure 2; Table 8).

At a high level of both environmental and health drivers, all message conditions had a significant conditional effect. The most influential conditions were loss messages and non-loss messages, followed by gain and non-gain. Therefore, participants strongly interested in both the environmental and health consequences of their food choices were sensitive to all message frames. However, they also showed a more evident preference for messages focusing on the environmental damages (loss condition) or avoidable damages (non-loss condition) derived from not purchasing or purchasing local food (Figure 2; Table 8).

A synthesis of the above analyses on the interaction between message conditions and food choice drivers in influencing participants’ choices is reported in Table 9. To simplify, we defined as “pro-environmental consumers”, the participants with high/medium level of environmental driver and low/medium level of health driver; as “healthy consumers”, the participants with high/medium level of health driver and low/medium level of the environmental driver; and as “oppositive consumers”, the participants with a low level of both environmental and health drivers. In sum, messages formulated in terms of the presence or absence of gains were more effective for participants who were pro-environmental consumers. Conversely, messages formulated in terms of the presence or absence of losses were more effective for participants who were healthy consumers. Instead, moderately pro-environmental and healthy consumers preferred simpler messages (gain or loss) over more complex messages (non-loss or non-gain), whereas strongly pro-environmental and healthy consumers were more persuaded by loss messages. Finally, weakly environmental and health consumers were oppositive; that is, they were not persuaded by any message.

## 6. Discussion

### 6.1. Prefactual Environmental Messages Influence Local Food Choice

The present study offers four main contributions to research aimed at investigating how to promote sustainable food choices. The first contribution regards the identification of key messages to increase the selection of local food. Our research shows that asking people to read prefactual messages about the environmental consequences of purchasing local food positively influences the choice of this food. These results extend previous empirical evidence on the usefulness of leveraging pro-environmental motivations to persuade consumers to change their eating habits [16,18,19], demonstrating its validity also in the case of local food purchases. This outcome is promising, given that, so far, the only study on the promotion of local foods—which tested the effectiveness of (non prefactual) messages focused on health, support for the community, and good quality—did not find any significant effect on consumers [20].

### 6.2. Environmental and Health Drivers of Food Choice Influence Message Frame Effectiveness

Our study shows that prefactual messages differing as to outcome sensitivity level (i.e., gain, non-loss, non-gain, and loss messages) have different effects according to the prevailing drivers of the recipients’ food choice. This result contributes to the debate regarding the utility of formulating the consequences of a recommended behavior as gain, non-loss, non-gain, or loss. As discussed in the theoretical background, previous literature collected contradictory results on the different impacts of differently framed environmental messages [27,29,30,31,32,33]. This can be partly due to the fact that the relative effectiveness of these messages depends on the psychosocial features of the receivers [37]. Consistently, we showed that the four message frames enhance the selection of local food depending on the receivers’ environmental and health drivers. This is in line with the notion that consumers of local foods are flexible consumers and can be guided by diverse drivers [49]. 

### 6.3. Ecological Intervention

The third contribution of the present study regards the methodology employed. In most previous studies aimed at promoting sustainable food choices, participants were exposed to a single message at once by asking them to read a message during a questionnaire completion. Differently, in our study we adopted a more ecological approach, exposing people to multiple messages sent via a smartphone app for two weeks. This is consistent with the fact that in ordinary life consumers are repeatedly exposed to advertisements or public promotional campaigns. Our ecological intervention (in real life and in real time) showed that digital communication, which can be easily provided on a large scale, can effectively convince people to select local foods. This result opens the way to the possibility of implementing campaigns that ensure the reach of a large number of recipients in a short time, reducing social inequalities in access to information that helps people protect their health and the environment.

### 6.4. Going beyond Intentions 

Finally, studies on this topic often measure general intentions related to food choice, such as to which extent the participant wants to buy local food in the following month. However, this measurement may refer to a conscious willingness that will not be necessarily translated into concrete action in the immediate future. This mismatch between intention and behavior may arise from other intervening factors. For instance, people may intend to buy local food, but other information could distract them from their willingness during the purchase, such as when they do not consider the product’s provenience because they are more attracted by other characteristics of the product (e.g., price or weight). To consider these possible distractors and have a more precise measurement of the intervention effect, in our study we created a food choice task as realistic as possible, asking people to choose different fruits to buy while receiving different information about their characteristics (not only provenience but also price, weight, category, and date of packaging). All these innovative aspects of our study allowed us to collect our data in a context that is closer to what we could observe in real purchasing contexts. 

### 6.5. Limitations 

Our research has several limitations. First, in the light of the existing gap between choice tasks regarding a certain behavior and its actual performance [50], the lack of measurement of the actual behavior is the most important limitation of the present study. Second, the choice task required only the selection of fruits; thus, our results may be generalized with caution to the selection of other local food products. Third, within our questionnaire, we did not measure participants’ ethnocentrism, which may influence their fruit selection. Finally, our sample was restricted to Italian people; thus, the data may not be generalized to other countries. Moreover, participants were exposed only for two weeks to the messages on the environmental outcomes of purchasing or not purchasing local food. Thus, we were able to assess only small and short-term effects. Messages delivered over a longer time span and with repeated exposure could yield larger and long-term effects on recipients’ attitudes and intentions [51,52].

### 6.6. Future Directions

Future research should test the cognitive and emotional processing involved in reading messages on environmental consequences formulated with different frames. Future studies could also verify whether the effects of gain, non-loss, non-gain and loss messages are the same when the presented outcomes are different from the ones presented here. We cannot exclude that there might be systematic differences among messages that propose the same behavior to obtain different outcomes (e.g., social sustainability). Similarly, a close consideration of how environmental messages focused on different behaviors (e.g., organic food purchase) may differently interact with food choice drivers would be useful. Future studies could also deepen our understanding of the effects of the four types of messages employed here, considering their fit with other individual characteristics not considered here, such as the rational or moral approach towards food purchasing [53], trust towards the health recommendation provided by public authorities [54,55], stage of change [56] and prior intentions [57].

### 6.7. Practical Implications

Considering the significant reduction in environmental impact deriving from the purchase of local food, selecting effective recommendation messages in this regard can usefully contribute to the success of environmental policies. In the current study, we found that a relatively simple and low-cost intervention such as the one we proposed can lead to a significant enhancement in self-reported choice of local food. Thus, the practical implications of our results include the possibility of using prefactual messages of the type employed in our study to increase the choice of local food. For example, these messages may be used to deliver recommendations via online communication within digital promotion campaigns. Institutions might also use their social media channel to send environmental messages to prompt sustainable food choices. 

More technological innovation and public policy efforts should also be devoted to engaging receivers in sustainable food choices by framing messages that accurately fit with their characteristics. To send such tailored messages to the right audience via social networks, public authorities could create chatbots able to automatically select and send different messages to different receivers. Collaboration between social psychology and artificial intelligence (AI) can help achieve this goal [58]. On the one hand, social psychology can develop consolidated models of the psychological factors that underlie food choices. On the other hand, artificial intelligence can, starting from psychosocial models, assess their predictive capacity, as well as simulate their application to larger populations that may have characteristics that differ from those of the initial sample on which the models were tested.

## 7. Conclusions

In our research, we found that repeated provision of prefactual messages (i.e., “If … then”) on the environmental consequences associated with local food purchase is effective in enhancing the selection of local food in a choice task, but not with people with a low level of health or environmental driver. In line with previous literature highlighting the role of environmental and health drivers in promoting sustainable food choices [39,40], we found that the effectiveness of messages differing as to the outcome sensitivity level of message framing (i.e., gain, non-loss, non-gain and loss) varied according to the level of the receivers’ environmental and health drivers. Pro-environmental consumers, were more convinced by messages formulated in terms of gains and non-gains, whereas healthy consumers were more convinced by messages formulated in terms of losses or non-losses. These results support the idea that tailored communication is the most effective strategy to encourage sustainable food choices. 

## Figures and Tables

**Figure 1 foods-11-01268-f001:**
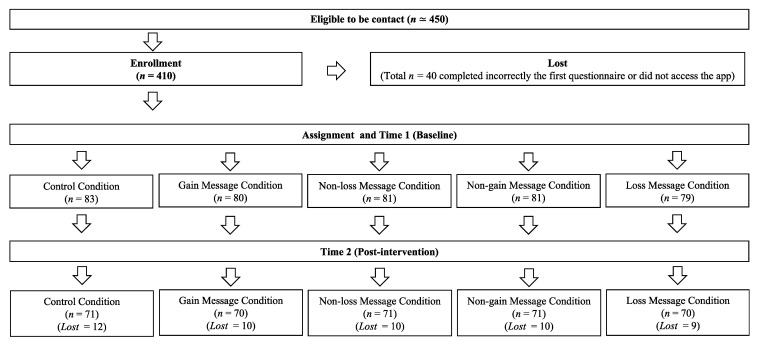
Flow chart of participants’ recruitment.

**Figure 2 foods-11-01268-f002:**
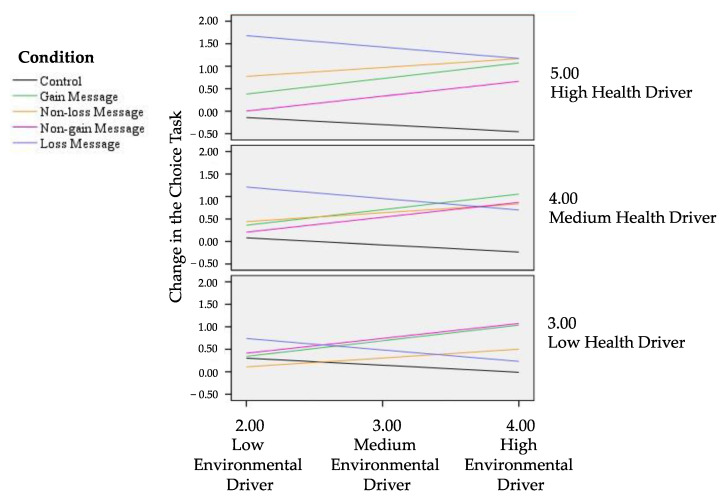
Change in the fruit choice task according to message framing conditions and participants’ levels of environmental and health drivers.

**Table 1 foods-11-01268-t001:** Demographics of the study sample.

Characteristic	Total Sample	
Gender		
Female	52.4%	
Male	46.7%	
Non-binary	0.9%	
Age		
Young (18–24 years)	50.7%	
Young Adults (25–35 years)	34.8%	
Adults (35–54)	12.5%	
Senior (55–65)	2.0%	
*M*		26.95
*SD*		7.87
Education		
Secondary School	2.3%	
High School Diploma	53.5%	
University Degree	44.2%	
Marital Status		
Single	76.2%	
Married	6.8%	
Cohabiting Couple	14.2%	
Separated/Divorced	0.3%	
Not declared	2.5%	
Monthly Income		
EUR < 1200	53.8%	
EUR 1200–2500	21.2%	
EUR > 2500	5.1%	
Not declared	19.8%	
Place of Residence		
Northern Italy	51.0%	
Central Italy	21.0%	
Southern Italy	19.0%	
Islands	9.0%	
Number of Residents in your Municipality		
Less than 10,000	18.4%	
Between 10,000 and 30,000	19.0%	
Between 30,000 and 100,000	23.5%	
Between 100,000 and 250,000	10.2%	
Between 250,000 and 500,000	4.0%	
More than 500,000	24.9%	

*M* = Mean; *SD* = Standard Deviation.

**Table 2 foods-11-01268-t002:** Means and standard deviations of measured variables in each message condition.

	Control Condition (*n* = 71)		Gain Message Condition (*n* = 70)		Non-Loss Message Condition (*n* = 71)		Non-Gain Message Condition (*n* = 71)		Loss Message Condition (*n* = 70)		Total (N = 353)	
Variables	*M*	*SD*	*M*	*SD*	*M*	*SD*	*M*	*SD*	*M*	*SD*	*M*	*SD*
Place Identity	4.47	1.69	4.36	1.69	4.08	1.61	4.12	1.86	3.95	1.71	4.20	1.69
Health Driver	3.63	0.99	3.83	0.92	3.63	1.06	3.90	0.99	3.69	1.03	3.74	1.00
Environmental Driver	3.13	0.97	2.86	1.03	3.14	1.03	3.39	1.10	3.07	1.11	3.20	1.07
Choice Task at Time 1	3.06	11.19	3.63	1.13	2.93	1.00	2.84	1.04	2.71	0.95	2.88	1.04
Choice Task at Time 2	3.04	1.17	4.64	1.11	3.48	1.13	3.51	1.14	3.50	1.16	3.43	1.16
Message Tone	-	-	5.60	1.33	5.42	1.09	5.08	1.45	4.54	1.54	5.16	1.41
Message Involvement	-	-	5.52	0.99	5.16	1.29	5.32	1.07	5.32	1.20	5.33	1.15
Message Trust	-	-	5.33	0.97	5.18	1.08	5.18	1.03	5.24	1.06	5.23	1.03
Systematic processing	-	-	5.15	0.90	4.94	1.36	5.05	1.17	5.02	1.09	5.04	1.14

Choice Task: selection of local fruit.

**Table 3 foods-11-01268-t003:** Correlations between measured variables at Time 1 and Time 2.

	1.	2.	3.	4.	5.	6.	7.	8.	9.
1. Place Identity	1								
2. Health Driver	0.13 *	1							
3. Environmental Driver	0.04	0.39 **	1						
4. Choice Task at Time 1	0.08	0.03	0.07	1					
5. Choice Task at Time 2	0.10	0.16 **	0.20 **	0.26 **	1				
6. Message Tone	0.15 *	0.14 *	0.21 **	0.02	0.12 *	1			
7. Message Involvement	0.25 **	0.23 **	0.27 **	0.03	0.22 **	0.47 **	1		
8. Message Trust	0.21 **	0.21 **	0.26 **	−0.02	0.15 *	0.37 **	0.54 **	1	
9. Systematic processing	0.17 **	0.29 **	0.34 **	0.02	0.26 **	0.42 **	0.73 **	0.55 **	1

* *p* < 0.05; ** *p* < 0.001.

**Table 4 foods-11-01268-t004:** Repeated measure ANOVA results with choice task as a dependent variable.

Predictor	Sum of Square	*df*	Mean Square	*F*	*p*	*ηp^2^*
Intercept	6994.77	1	6994.77	4520.53	0.00	0.93
Time	53.55	1	53.55	61.53	0.00	0.15
Condition	3.39	4	0.85	0.55	0.70	0.01
Time × Condition	15.25	4	3.81	4.39	0.00	0.05
Error (Time)	301.14	346	0.87			
Error (Group)	535.38	346	1.55			

**Table 5 foods-11-01268-t005:** Moderation model results with choice task as a dependent variable.

	*B*	*SE*	*t*	*p*	95% *CI*	
					*LL*	*UL*
**Environmental Driver**						
Gain Message Condition	0.50	0.22	2.23	0.02	0.61	0.94
Non-Loss Message Condition	0.35	0.23	1.53	0.13	−0.10	0.81
Non-Gain Message Condition	0.49	0.22	2.19	0.03	0.05	0.93
Loss Message Condition	−0.10	0.23	−0.43	0.67	−0.54	0.35
**Health Driver**						
Gain Message Condition	0.24	0.25	0.97	0.33	−0.25	0.73
Non-Loss Message Condition	0.55	0.23	2.45	0.01	0.11	1.00
Non-Gain Message Condition	0.02	0.23	0.08	0.94	−0.44	0.47
Loss Message Condition	0.69	0.23	2.98	0.00	0.23	1.15

*CI* = confidence interval; *LL* = lower limit; *UL* = upper limit.

**Table 6 foods-11-01268-t006:** Conditional effects of message conditions at values of the environmental driver.

	*B*	*SE*	*t*	*p*	95% *CI*	
					*LL*	*UL*
**Low Environmental Driver–Low Health Driver**						
Gain Message Condition	0.04	0.34	0.12	0.91	−0.63	0.61
Non-Loss Message Condition	−0.20	0.33	−0.58	0.56	−0.85	0.46
Non-Gain Message Condition	0.11	0.35	0.31	0.76	−0.58	0.80
Loss Message Condition	0.44	0.32	1.35	0.18	−0.20	1.07
**Low Environmental Driver–Medium Health Driver**						
Gain Message Condition	0.28	0.37	0.76	0.45	−0.45	1.01
Non-Loss Message Condition	0.36	0.37	0.97	0.33	−0.37	1.09
Non-Gain Message Condition	0.13	0.37	0.34	0.73	−0.60	0.86
Loss Message Condition	1.13	0.36	3.14	0.00	0.42	1.84
**Low Environmental Driver–High Health Driver**						
Gain Message Condition	0.54	0.27	1.97	0.05	0.00	1.08
Non-Loss Message Condition	0.16	0.25	0.62	0.53	−0.34	0.66
Non-Gain Message Condition	0.60	0.27	2.17	0.03	0.06	1.14
Loss Message Condition	0.34	0.26	1.30	0.19	−0.17	0.85

*Note.**CI* = Confidence Interval; *LL* = Lower Limit; *UL* = Upper Limit.

**Table 7 foods-11-01268-t007:** Conditional effects of message conditions at values of the moderator.

	*B*	*SE*	*t*	*p*	95% *CI*	
					*LL*	*UL*
**Medium Environmental Driver–Low Health Driver**						
Gain Message Condition	0.54	0.27	1.97	0.05	0.00	1.08
Non-Loss Message Condition	0.16	0.25	0.62	0.53	−0.34	0.66
Non-Gain Message Condition	0.60	0.27	2.17	0.03	0.06	1.14
Loss Message Condition	0.34	0.26	1.30	0.19	−0.17	0.85
**Medium Environmental Driver–Medium Health Driver**						
Gain Message Condition	0.78	0.24	3.31	0.00	0.32	1.25
Non-Loss Message Condition	0.71	0.24	3.01	0.00	0.25	1.18
Non-Gain Message Condition	0.61	0.24	2.59	0.01	0.15	1.08
Loss Message Condition	1.03	0.23	4.39	0.00	0.57	1.49
**Medium Environmental Driver–High Health Driver**						
Gain Message Condition	1.02	0.40	2.56	0.01	0.24	1.81
Non-Loss Message Condition	1.27	0.39	3.28	0.00	0.51	2.03
Non-Gain Message Condition	0.63	0.38	1.66	0.10	−0.11	1.38
Loss Message Condition	1.72	0.39	4.45	0.00	0.96	2.49

*CI* = Confidence Interval; *LL* = Lower Limit; *UL* = Upper Limit.

**Table 8 foods-11-01268-t008:** Conditional effects of message conditions at values of the moderator.

	*B*	*SE*	*t*	*p*	95% *CI*	
					*LL*	*UL*
**Medium Environmental Driver–Low Health Driver**						
Gain Message Condition	1.05	0.37	2.82	0.00	0.32	1.78
Non-Loss Message Condition	0.51	0.36	1.44	0.15	−0.19	1.21
Non-Gain Message Condition	1.08	0.37	3.06	0.00	0.39	1.78
Loss Message Condition	0.24	0.37	0.66	0.51	−0.48	0.96
**Medium Environmental Driver–Medium Health Driver**						
Gain Message Condition	1.29	0.27	4.69	0.00	0.75	1.83
Non-Loss Message Condition	1.07	0.28	3.75	0.00	0.51	1.63
Non-Gain Message Condition	1.10	0.27	4.08	0.00	0.57	1.63
Loss Message Condition	0.93	0.29	3.23	0.00	0.36	1.50
**Medium Environmental Driver–High Health Driver**						
Gain Message Condition	1.53	0.37	4.12	0.00	0.80	2.26
Non-Loss Message Condition	1.62	0.37	4.37	0.00	0.89	2.35
Non-Gain Message Condition	1.12	0.36	3.12	0.00	0.41	1.82
Loss Message Condition	1.63	0.37	4.34	0.00	0.89	2.36

*Note. CI* = Confidence Interval; *LL* = Lower Limit; *UL* = Upper Limit.

**Table 9 foods-11-01268-t009:** Best message framing as a function of the interaction effects emerged from moderation analysis.

Type of Consumer	Environmental Driver	Health Driver	Best Framing	Worst Framing
Oppositive Consumer *	Low	Low	None	None
Pro-environmental Consumer	Medium	Low	Gain and non-gain	Non-loss and loss
Pro-environmental Consumer	High	Low	Gain and non-gain	Non-loss and loss
Pro-environmental Consumer	High	Medium	Gain and non-gain	None
Healthy Consumer	Low	Medium	Loss	Gain, non-loss, non-gain
Healthy Consumer	Low	High	Loss	Gain, non-loss, non-gain
Healthy Consumer	Medium	High	Loss	Non-gain
Pro-environmental and Healthy Consumer	Medium	Medium	Loss and gain	None
Pro-environmental and Healthy Consumer	High	High	Loss	None

* Oppositive consumer: participants not persuaded by any messages.

## Data Availability

The data presented in this study are available on request from the corresponding author.

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
