# Peer review of "The Influence of Message Framing on Consumers’ Selection of Local Food"

_foods, 2022, doi:10.3390/foods11091268_

Round 1

Reviewer 1 Report

Well-written paper with an interesting impact of messaging towards food choice, however - there are quite a few things that are missing that is needed before the review can be carried out fully

The authors concluded that 80 would be the ideal number it seems based on their groupings. Were there any prelim power analysis that was carried out?

Can the authors include the statistical approach in their materials and methods?

So the authors attempted some segmentation, is it possible to include the info on how this was done?

How did the authors arrive in type of cosumers and where does the best-worst framing is from?

Lots of information on how things are done is missing, ths needs to be added.

Is it possible for the authors to summarise their key highlights and use them to subsection their Discussion?

Author Response

Well-written paper with an interesting impact of messaging towards food choice, however - there are quite a few things that are missing that is needed before the review can be carried out fully

1) The authors concluded that 80 would be the ideal number it seems based on their groupings. Were there any prelim power analysis that was carried out?

1) Yes, to estimate the sample size we ran a power analysis reported on p. 4, lines 175-180.

2) Can the authors include the statistical approach in their materials and methods?

2) Thanks for rising this point, we have now included the statistical approach in the Data Analysis section (p. 7, lines 302-316).

3) So the authors attempted some segmentation, is it possible to include the info on how this was done?

3) As regards segmentation, we have categorized participants according to their levels of environmental and health drivers. We have now better clarified this aspect on p. 15 (lines 503-508).

4) How did the authors arrive in type of cosumers and where does the best-worst framing is from?

4) In the Results section, we have now better explained how we defined the types of consumer according to the results of our moderation analysis (p. 15, lines 503-508).

5) Lots of information on how things are done is missing, ths needs to be added.

5) As suggested, we have now extensively explained the study procedure: Data collection through the PsyMe app and Prolific (p. 5, lines 181-201), and data analysis (p. 7, lines 302-316).

6) Is it possible for the authors to summarise their key highlights and use them to subsection their Discussion?

6) We have now organized the Discussion in subsections as suggested (p. 15, lines 522-629).

Reviewer 2 Report

Interesting topic assessed with a valid approach. However, several remarks should be addressed.

The literature review assessing the case of local foods and short food supply chains does not cover recent publications and the mixed results of their impact on environmental sustainability are not covered by the manuscript.

I don't see the logic of having one hypothesis and two research questions.

The process of data collection and sampling is not clear enough. Please describe in more detail, e.g., how prolific works. Also, there is only very limited data provided on the app used for the project.

It is not validated, why only fruits are included in the research, e.g., why fresh vegetables are not included.

How does your model handle the moderating effect of having a non-local alternative (outside of the 100 km radius) not from Italy, not from the EU, or not from Europe? The ethnocentricity of the consumers, per sé, might have a substantial influence on their decisions.

In the present form, the Results part is very detailed and hard to follow, as many non-significant results are also displayed in detail. On the contrary, part Conclusions is too short and fails in concluding the most important findings of the manuscript. 

Please check the misspellings (e.g. Nord Italy in line 271).

Author Response

Interesting topic assessed with a valid approach. However, several remarks should be addressed.

7) The literature review assessing the case of local foods and short food supply chains does not cover recent publications and the mixed results of their impact on environmental sustainability are not covered by the manuscript.

7) We have now better discussed the mixed results about the environmental impact of the local food chains (p. 1, lines 43-58).

8) I don't see the logic of having one hypothesis and two research questions.

8) As stated on p. 4 (lines 160-163), our study has mainly an exploratory aim due to the lack of previous studies on the promotion of local food. For this reason, we have only hypothesized an effect of being exposed to environmental messages as compared to not being exposed to them, and we have formulated two research questions regarding the effect of message framing.  

9) The process of data collection and sampling is not clear enough. Please describe in more detail, e.g., how prolific works. Also, there is only very limited data provided on the app used for the project.

9) We have now described in more detail how Prolific works and provided more details about the PsyMe app (p. 5, lines 181-201).

10) It is not validated, why only fruits are included in the research, e.g., why fresh vegetables are not included.

10) In the Measures at Time 1 section we have now better explained why we included only fruits in the experiment (p. 5, lines 228-230).

11) How does your model handle the moderating effect of having a non-local alternative (outside of the 100 km radius) not from Italy, not from the EU, or not from Europe? The ethnocentricity of the consumers, per sé, might have a substantial influence on their decisions.

11) All participants selected fruit from a list that featured the same number of local, European, and non-European products. For this reason, we did not consider a possible moderation of being exposed to products with different origins. To control for the effect of participants’ attachment to their origin, we have now included an additional analysis with the place identity variable as a covariate (Appendix B, p. 23; see also #8 above). We have also mentioned the lack of the ethnocentricity measurement in the Limitations section (p. 16, lines 584-586).

12) In the present form, the Results part is very detailed and hard to follow, as many non-significant results are also displayed in detail. On the contrary, part Conclusions is too short and fails in concluding the most important findings of the manuscript. 

12) As suggested, we have now omitted the description of non-significant results in the Results section. We have also expanded the Conclusions by discussing the implications of our findings in more detail (p. 631-643).

13) Please check the misspellings (e.g. Nord Italy in line 271).

13) We have fixed the misspellings throughout the text.

Round 2

Reviewer 1 Report

I'd like to thank the authors for revising their manuscript.

Author Response

Thank you for this suggestion. We have now checked the language and the spell of the whole manuscript.

Reviewer 2 Report

The manuscript has been improved.

However, some issues are still in need of improvement:

  • Prolific is still not explained well (what kind of webpage is it, how it can be used for research etc.)
  • The rationale behind excluding vegetables and focusing only on fruits is unclear

Author Response

Please, find below a detailed response to all the points the reviewer indicated, with the original comments in italics.

The manuscript has been improved.

However, some issues are still in need of improvement:

2) Prolific is still not explained well (what kind of webpage is it, how it can be used for research etc.)

2) As suggested, we have now better described the Prolific platform (p. 4, lines 186-200).

3) The rationale behind excluding vegetables and focusing only on fruits is unclear

3) Thank you for giving us the opportunity to better explain it. We clarified the rationale behind this choice on p. 6 (lines 244-249).